# Sensors in Bone: Technologies, Applications, and Future Directions

**DOI:** 10.3390/s24196172

**Published:** 2024-09-24

**Authors:** Afreen Anwar, Taruneet Kaur, Sachin Chaugule, Yeon-Suk Yang, Aryan Mago, Jae-Hyuck Shim, Aijaz Ahmad John

**Affiliations:** 1Department of Biology and Biotechnology, Worcester Polytechnic Institute, 100 Institute Road, Worcester, MA 01609, USA; 2Department of Biotechnology and Zoology, Baba Ghulam Shah Badshah University, Rajouri 185234, India; 3Faculty of Engineering and Design, Carleton University, 125 Colonel By Dr, Ottawa, ON K1S 5B6, Canada; 4Department of Medicine, Division of Rheumatology, University of Massachusetts Chan Medical School, Worcester, MA 01655, USA; 5Horae Gene Therapy Center, University of Massachusetts Chan Medical School, Worcester, MA 01655, USA; 6Li Weibo Institute for Rare Diseases Research, University of Massachusetts Chan Medical School, Worcester, MA 01655, USA

**Keywords:** bone, osteoporosis, osteoblasts, osteoclasts, biosensors, biomechanical sensors, bone biomarkers

## Abstract

Osteoporosis, a prevalent ailment worldwide, compromises bone strength and resilience, particularly afflicting the elderly population. This condition significantly heightens susceptibility to fractures even from trivial incidents, such as minor falls or impacts. A major challenge in diagnosing osteoporosis is the absence of discernible symptoms, allowing osteoporosis to remain undetected until the occurrence of a fracture event. Early symptom detection and swift diagnosis are critical for preventing severe issues related to bone diseases. Assessing bone turnover markers aids in identifying, diagnosing, and monitoring these conditions, guiding treatment decisions. However, conventional techniques for measuring bone mineral density are costly, time-consuming, and require specialized expertise. The integration of sensor technologies into medical practices has transformed how we monitor, diagnose, and treat various health conditions, including bone health and orthopedics. This review aims to provide a comprehensive overview of the current state of sensor technologies used in bone, covering their integration with bone tissue, various applications, recent advancements, challenges, and future directions.

## 1. Introduction

Bone, a highly dynamic tissue, undergoes continuous remodeling throughout life, even after reaching maturity. Bone remodeling is a process driven by the coordinated actions of osteoclasts and osteoblasts [1]. This process is driven by mechanical stimuli, where osteoclast-mediated resorption of old bone is followed by the formation of new bone by osteoblasts [2]. This homeostatic equilibrium of bone resorption and formation is crucial for healthy bone, but disruptions like hormonal changes in menopause and aging can impact bone remodeling leading to osteoporosis [3]. Osteoporosis encompasses a spectrum of conditions marked by reduced bone mass, compromised mineralization, diminished strength, structural deterioration, and susceptibility to nontraumatic fracture [4]. However, patients with osteoporosis do not typically present with noticeable symptoms; thus, the disease often goes unnoticed until a fracture occurs. Detecting bone biomarkers is crucial for promptly diagnosing ailments like osteoporosis, bone cancer, and infections, as they signify underlying processes.

Biomarkers for bone health include cells, enzymes, hormones, and gene products that help identify different stages of disease and facilitate effective treatment. Precise identification and monitoring of specific bone biomarkers are instrumental in refining a diagnosis and optimizing therapeutic interventions for bone diseases [5]. In addition, early symptom detection and a rapid diagnosis are crucial for averting serious complications related to bone diseases. The assessment of bone turnover markers plays a pivotal role in pinpointing, diagnosing, and tracking these conditions, informing treatment choices. Nonetheless, conventional methods used to measure bone mineral density are expensive, time-intensive, and demand specialized skills [6]. Therefore, there is a need for portable electronic devices to convert biological information into a readable output, yet introducing these devices within a biological environment, e.g., within bones, poses challenges due to complexities in attaching the device and signal processing.

Unlike conventional methods, which fail to provide real-time assessments of bone biomechanics, bone biomechanical sensors offer immediate feedback on bone status. This advantage highlights the capability of biomechanical sensors to detect issues as they occur, setting them apart from existing technologies [7]. The incorporation of sensor technologies into medical care has revolutionized the detection, diagnosis, and treatment of numerous health conditions, including those related to bone health, where sensors stand out as an especially promising frontier. Current biosensors encompass wearable or implantable devices enabling communication via USB (wearable) or wireless methods like Bluetooth or Wi-Fi (for both wearables and implants) [8]. Recent progress in materials science and biomedical engineering has led to the development of biocompatible sensors that can be implanted within bones or attached to their surfaces. These sensors can track various physiological parameters, such as strain, pressure, and alterations in chemical composition. By doing so, they offer valuable insights into bone health, the progression of healing, and the early detection of potential issues before they escalate. Bone sensors can be divided into physical sensors and biosensors. Physical sensors measure parameters like temperature, strain, and pressure, converting them into readable signals. Biosensors, comprising biorecognition domains, transducers, and signal read-out systems, detect chemical substances.

## 2. An Overview of the Available Bone Turnover Biomarkers and Existing Biosensors for Monitoring Bone Health

Bone turnover biochemical markers (BTBMs) are substrates of bone-tissue enzymes or bone-associated proteins produced during bone remodeling. They can be measured using blood or urine samples and provide insight into bone remodeling activity [9,10]. BTBMs are categorized into two main groups: markers of bone formation and markers of bone resorption. Commonly used analytical methods to measure BTBMs are summarized in Table 1.

### 2.1. Biomarkers of Bone Formation

#### 2.1.1. Alkaline Phosphatase (ALP)

ALPs are enzymes localized in the cell membranes of osteoblasts. These enzymes exist in various isoforms, originating from different tissues, such as the liver, bone, intestine, and kidney [11,12,13]. Osteoblasts secrete bone-specific ALP (BALP) during bone formation, which serves as a crucial biomarker for this process. The clinical significance of BALP measurement lies in its high specificity, making it increasingly favored [14,15,16]. BALP measurement assays are widely used for the clinical evaluation of osteoporosis treatments due to their prevalence, accessibility, and common usage [17,18,19].

#### 2.1.2. Osteocalcin (OC)

Osteocalcin is a small noncollagenous protein containing vitamin K and glutamic acid residues, synthesized by osteoblasts and odontoblasts [20,21]. It is considered a specific biomarker of osteoblast activity [22,23]. Upon release from osteoblasts, most newly produced OC integrates into the bone matrix, with a small fragment entering the circulation that is measurable by immunoassays [24,25].

#### 2.1.3. Propeptides of Type I Procollagen (PICP and PINP)

Propeptides of type I procollagen, including amino-terminal (PICP) and carboxy-terminal (PINP) peptides, are biomarkers of bone formation. They are extracted from type I collagen, which constitutes a significant portion of the bone’s organic matrix. These peptides are divided and released into the circulation. Therefore, they serve as indicators of bone formation [26].

### 2.2. Biomarkers of Bone Resorption

Most bone resorption markers, except for tartrate-resistant acid phosphatase (TRACP), are derived from bone collagen. Recent studies have explored noncollagenous markers of bone resorption, including bone sialoprotein and markers derived from osteoclasts.

#### 2.2.1. Hydroxyproline (OHP)

Hydroxyproline is a crucial amino acid formed by the post-translational hydroxylation of proline and is a byproduct of collagen degradation. Bone-derived OHP is primarily broken down into free amino acids, subsequently processed by the kidney and metabolized by the liver, with only 10–15% excreted in urine. The majority of OHP exists in a peptide form (~90%), with a smaller fraction in a free form, and the remainder in a polypeptide form [27,28]. OHP has been replaced by specific bone resorption markers due to nonspecificity, dietary restrictions (particularly gelatin containing foods), and contributions from newly synthesized collagen degradation [29].

#### 2.2.2. Hydroxylysine-Glycosides

Hydroxylysine-glycosides are amino acids found in collagen, specifically associated with bone. These amino acids are formed during the post-translational processing of collagen. Unlike OHP, they are not affected by diet [30]. Hydroxylysine-glycosides exist in two forms: glycosyl-galactosyl-hydroxylysine (GGHL) and galactosyl-hydroxylysine (GHL) [31,32]. As collagen breaks down, GGHL and GHL are released into the circulation and can be measured in urine [33]. However, due to their biological variability, hydroxylysine-glycosides are considered a less reliable marker of bone resorption [34].

#### 2.2.3. Collagen Crosslink Molecules

Collagen crosslink molecules, including pyridinoline (PYD) and deoxypyridinoline (DPD), form during the extracellular maturation of collagen. These crosslink molecules are released into circulation during the process of bone resorption. They play a crucial role in strengthening the collagen molecule mechanically [35]. PYD is found in bone, cartilage, vessels, and ligaments, while DPD is predominantly found in bone and dentin, with limited presence in skin and other sources. DPD is considered a more sensitive marker than PYD. Due to the higher rate of bone turnover compared to cartilage, vessels, and ligaments, PYD and DPD present in urine and serum are predominantly derived from bone, making them among the most reliable biomarkers of bone resorption [36,37].

#### 2.2.4. Cross-Linked Telopeptides of Type I Collagen

Cross-linked telopeptides of type I collagen are derived from the amino-terminal (N-terminal) and the carboxy-terminal (C-terminal) of type I collagen and have, thus, been named NTX-I and CTX-I, respectively [38,39]. Both NTX-1 and CTX-1 are widely used markers to assess bone resorption in serum and urine [40]. Recent investigations have demonstrated that immunoassays, such as the enzyme-linked immunosorbent assay (ELISA) and the electrochemiluminescence immunoassay (ECLIA), provide higher sensitivity for urine NTX-I and serum CTX-I over DPD in monitoring antiosteoclastic therapies [41,42].

#### 2.2.5. Bone Sialoprotein (BSP)

Bone sialoprotein is a phosphorylated glycoprotein, accounting for 5–10% of the noncollagenous bone matrix [43]. It is found in mineralized tissues, like bone and dentine, as well as various cells, including osteoblasts, odontoblasts, osteoclast-like cells, and cancerous cell lines [43,44]. BSP plays a significant role in cell-matrix adhesion procedures and can be used to assess osteoclast-mediated bone resorption [45].

#### 2.2.6. Tartrate-Resistant Acid Phosphatase (TRAP)

Tartrate-resistant acid phosphatases (TRAPs) are a group of enzymes classified as acid phosphatases [46]. These enzymes are released into circulation in two forms, TRAP5a and TRAP5b, which share similar structures but differ in their optimum pH and carbohydrate content. TRAP5a is derived from macrophages, while TRAP5b is produced by osteoclasts [30,47,48]. TRAP5b contributes to bone matrix degradation, making it a valuable marker for studying osteoclast resorption activity [49].

#### 2.2.7. Cathepsin K

Cathepsin K is a member of the cysteine protease group that can cleave both the helical and telopeptide parts of type I collagen [50,51]. Cathepsin K is secreted from osteoclasts and plays a crucial role in bone resorption [52]. Despite its potential as a biomarker of bone resorption, further research is necessary before its commercial application can be considered [53].

### 2.3. Analytical Methods for the Measurement of Bone Turnover Markers

Currently, ELISA, ECLIA, radioimmunoassay (RIA), and high-performance liquid chromatography (HPLC) are the most extensively used analytical methods for detecting and measuring BTBMs.

#### 2.3.1. Enzyme-Linked Immunosorbent Assay (ELISA)

ELISA is a powerful technique used for detecting and quantifying low concentrations of antigens in biological samples. It relies on enzymatic reactions to produce a color change that indicates the presence of the target molecule [54,55]. It is a well-known application for detecting peptides and proteins [56]. ELISA can be homogenous or heterogeneous. The homogenous method does not involve a washing step, making it easy to use, but is more costly and less sensitive. On the other hand, the heterogeneous method, involving a washing step to remove unbound materials, is more popular due to its intensified sensitivity [54,57]. Four different types of ELISA (direct, indirect, sandwich, and competitive) have been developed to improve the specificity in measuring various substrates, thus enhancing the overall measurement accuracy.

#### 2.3.2. Electrochemiluminescence Immunoassay (ECLIA)

ECLIA is a highly sensitive technique used to detect and quantify substances in a sample. This approach uses a combination of electrochemical and luminescence processes to achieve this. ECLIA relies on the interaction between an antibody and its antigen, producing a luminescent signal to quantify the target molecules [58].

#### 2.3.3. Radioimmunoassay (RIA)

RIA is a highly sensitive method for detecting antigens and antibodies [59]. It is based on the competitive binding of radiolabeled antigens and unlabeled antigens to an antibody. The antibody cannot distinguish between labeled and unlabeled antigens, leading to competition for binding sites. With an increased concentration of unlabeled antigens, more labeled antigens are displaced from the binding sites. The antigen concentration in the test sample is determined by estimating the reduction in the concentration of the radiolabeled antigen attached to the antibody in the test sample [59]. Gamma and beta-emitting isotopes are commonly used to label the antigens. While RIA is more sensitive and reliable than ELISA, the use of radioactivity makes it expensive and potentially hazardous to human health and the environment [60,61].

#### 2.3.4. High-Performance Liquid Chromatography (HPLC)

HPLC is a powerful technique used for quantitative and qualitative separation of single or multiple particles in biological and pharmaceutical samples. It is based on a column with a stationary phase, an injector for sample introduction into the mobile phase, a pump to drive the mobile phase through the column, and a detector to indicate particle retention time. The retention time, the time it takes for a molecule to elute from the column, is unique to each particle. Common detectors include UV-spectroscopy, mass spectrometry, electrochemical impedance spectroscopy, and fluorescence spectrometry [62,63].

HPLC can be classified into various types based on the stationary phase system, such as normal phase, reverse phase, size-exclusion, and ion-exchange HPLC. The system consists of a pump, injector, column, detector, and data acquisition system, and the HPLC column plays a crucial role in the separation process.

## 3. Biosensors for Bone Disorders

A biosensor refers to an analytical instrument that combines a biological recognition component with a transducer, aiming to identify and measure a distinct biological or chemical target. A biosensor functions as a device detecting biological or chemical reactions, producing signals directly proportional to the concentration of an analyte within the reaction [64]. A typical biosensor consists of (a) an analyte, (b) a bioreceptor, (c) a transducer, (d) an electronic component, and (e) a display (Figure 1).

An analyte is a target substance requiring detection, e.g., glucose levels in blood, whereas a bioreceptor is a biological recognition element (enzymes, cells, aptamers, DNA, antibodies, etc.) that selectively engages with the specific analyte of interest, initiating a detectable signal, such as light, heat, pH, charge, or mass change). This biorecognition event is converted into a measurable signal by the transducer. Most transducers generate either optical or electrical signals, typically in proportion to the extent of analyte–bioreceptor interactions. Further, the electronic component of a biosensor manages the transduced signal, undergoing signal conditioning, including amplification and conversion from analog to digital format. The processed signals are subsequently quantified by the biosensor’s display unit. The display, often a computer’s liquid crystal display or a direct printer, interprets data into comprehensible numbers or curves. It integrates hardware and software to present biosensor results in user-friendly formats, including numeric, graphic, tabular, or image-based outputs, customized to meet end-user preferences [65,66]. Biosensors are used in extensive applications across diverse domains, such as medical diagnostics, environmental surveillance, and food safety [67]. This is attributed to their capacity for carrying out precise, fast, and sensitive identification of substances [68].

**Figure 1 sensors-24-06172-f001:**
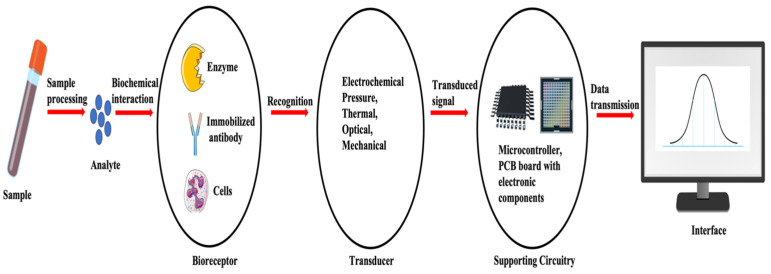
Basic components and flow of a biosensor system. Adapted and modified from [66,68].

Recently, there has been notable progress in biosensor techniques, enhancing the evaluation of both the biomechanical status and metabolic characteristics of bones [69]. These developments and research initiatives have enhanced the effectiveness of existing biosensors, leading to the development of cost-effective, dependable, and precise point-of-care devices. These biosensors play a crucial role in monitoring bone health by enabling the ongoing and timely evaluation of factors, like changes in bone mineral density, fluctuations in various proteins, and fracture alterations [70]. Interestingly, the field of biosensors is witnessing the evolution of diverse technologies capable of detecting bone cells and determining the concentration of BTBMs in biological samples. The process of creating a bone biosensor comprises three distinct phases: (a) selecting transducers, (b) constructing a sensing interface with recognition elements, and (c) conducting quantitative measurements using signal amplification and transduction elements [71]. Nanomaterials offer an enticing route for biosensor innovation. Biosensors utilizing gold nanoparticles, nanorods, nanowires, nanotubes, carbon dots, and quantum dots exhibit substantial promise in diagnostics due to their distinctive features, such as heightened electrical conductivity and extensive surface area, resulting in enhanced sensitivity [72]. Notably, there are different types of biosensors, such as electrochemical, colorimetric, and fluorescence biosensors, among others. These biosensor varieties are classified according to the methods they employ for signal read-out [73,74]. Some of the existing physical sensors and biosensors for bone health monitoring are mentioned in Table 2.

### 3.1. Electrochemical Biosensors

Electrochemical biosensors are analytical instruments that convert biochemical occurrences, like enzyme–substrate reactions and antigen–antibody interactions, into electrical signals, such as current, voltage, or impedance. Central to this sensor is the electrode, serving as a stable platform for anchoring biomolecules and facilitating electron transfer [99,100]. In 2017, Afsarimanesh and colleagues introduced a label-free biosensing method for monitoring CTX-I concentration in serum. They engineered artificial antibodies for CTX-I molecules, a prominent biochemical bone turnover marker known for providing early insights into the risk of osteoporotic fractures, using molecular imprinting. The sensor was then modified with the synthesized molecular imprinted polymer to enhance selectivity for the CTX-I biomarker. Serum samples from sheep blood were effectively analyzed using the proposed biosensor and demonstrated a strong correlation with ELISA, confirming the effectiveness of this innovative technique [89].

Further, a novel immunosensor was devised by Ramanathan et al., 2016, employing carbon nanotube (CNT) electrodes coated with gold nanoparticles to detect CTX, potentially revolutionizing bone metabolism detection and prognostics with its affordability and rapidity. Electrochemical impedance spectroscopy (EIS) was utilized to track the antigen–antibody binding events occurring on the gold-deposited CNT electrode surface. Type I CTX served as the model protein for evaluating the sensor’s efficacy, with detection capabilities reaching as low as 0.05 ng/mL [90]. Similarly, Yun et al. engineered a label-free immunosensor to target CTX-1. Their innovative approach involved employing self-assembled monolayers of dithiodipropionic acid on gold electrodes, facilitating streptavidin immobilization and subsequent binding of a biotinylated antibody. Utilizing EIS, the sensor achieved a detection limit of 50 ng/mL and a dynamic range of up to 3 μg/mL, significantly reducing analysis time to just four hours with a simplified single-step process, compared to traditional ELISA methods (Figure 2) [91]. Inal et al. developed a biosensor targeting osteocalcin for osteoporosis prognosis, utilizing covalent immobilization to affix an antiosteocalcin antibody onto a gold electrode. Characterized via cyclic voltammetry and impedance spectroscopy, the biosensor detected osteocalcin concentrations within 45 min, spanning 10 to 60 pg/µL [92]. Sappia et al. introduced an electrochemical biosensor for ALP determination, offering clinical utility with just 10 µL of serum [93].

Studies performed on the bone health of astronauts has provided vital information on bone health and advances for biosensors. Microgravity-induced bone mass reduction leads to osteopenia/osteoporosis in astronauts in space [101]. Osteocytes regulate bone remodeling by translating mechanical stimuli into biochemical signals that affect osteoblast and osteoclast functions. In microgravity (reduced mechanical loading), osteocytes increase sclerostin, blocking Wnt binding, decreasing osteoblast activity, and increasing osteoclast activity [102,103]. To address this, printed electrochemical biosensors have been developed for astronaut point-of-care testing. These sensors combine thin film material printing and 3D printing, incorporating carbon nanotubes, gold nanoparticles, silver nanoparticles, and dielectric inks to create traditional electrochemical sensor devices. They detect space osteopenia/osteoporosis by indirectly monitoring changes in bone mass through the biomarker NTX. Specific NTX antibodies are infused into the sensor’s electrode ink during manufacturing. This yields rapid, quantitative measurements using small handheld electronics. Ground and flight physicians gain frequent, real-time insights into space osteopenia/osteoporosis development, guiding appropriate countermeasures. Such innovation enhances astronauts’ bone health monitoring, critical for extended space missions [94].

### 3.2. Colorimetric Biosensors

Colorimetric biosensors are devices that identify and quantify biological or chemical substances by detecting alterations in color. In 2021, Samy et al. engineered a highly precise colorimetric sensor for ALP with a dynamic linear range spanning 0.5–225 U/L and a lower limit of detection (LOD) of 0.24 U/L. This sensor demonstrates exceptional accuracy, detecting ALP in human serum with a precision of 99.2 ± 1.5%. This colorimetric sensor for ALP relies on its enzymatic activity to convert p-aminophenol phosphate (pAPP) to pAP. In a solution with AgNPs and Ag+ ions, pAP facilitates AgNP growth by reducing Ag+ concentration. This method quantitatively detects ALP via pAP-mediated AgNP growth [95]. Likewise, Wignarajah et al. developed a colorimetric assay for detecting common biomarkers of periodontitis utilizing a magnetic nanoparticle biosensor [104].

### 3.3. Fluorescence Biosensors

Fluorescence biosensors are instruments used to identify and measure biological or chemical substances by detecting the fluorescent light emitted when they are excited by a particular wavelength of light [105]. In 2018, Park et al. introduced novel near-infrared (NIR) fluorescent probes (NIR-Phos-1, NIR-Phos-2) for highly sensitive ALP detection. ALP cleaves the phosphate group, altering NIR probe properties and leading to a significant fluorescent signal increase. The assay detects ALP activity from 0 to 1.0 U/mL with an LOD of 10^−5^−10^−3^ U/mL within 1.5 min. Real-time monitoring in live cells and animals showcases the probes’ potential. In vivo, ALP detection utilizes NIR probe-labeled 3D calcium-deficient hydroxyapatite scaffolds. Implanted mice exhibit ALP signal changes, suggesting early-stage ALP detection during neo-bone formation within bone defects [96]. Similarly, in 2019, Li Hou devised a selective, smartphone-based approach for visually detecting ALP, utilizing the unique attributes of amino-functionalized copper (II)-based metal–organic frameworks with oxidase mimicry and fluorescence capabilities [97].

### 3.4. Multiplex Assays

In recent developments, several multiplex immunoassays have emerged to concurrently identify multiple biomarkers, enhancing the accuracy of monitoring bone-remodeling processes [97]. Claudon et al. pioneered an automated multiplex assay for bone turnover markers, enabling the simultaneous measurement of CTX-I, PINP, OC, and parathyroid hormone (PTH) in just 20 μL of serum. This automated multiplex immunoassay demonstrated equivalent analytical precision and superior sensitivity compared to individual assays and is particularly advantageous in situations with limited sample volumes [97]. Another multiplex assay, Osteokit, was introduced by Khashayar et al. for bone marker assessment. They utilized a microfluidic platform to simultaneously measure OC and CTX-I in serum. Their results indicated comparable sensitivity between Osteokit and the conventional method, ECLIA, with a total assay time reported to be only 10 min, shorter than the time required by ECLIA [98].

### 3.5. Label-Free Biosensors

Most of the techniques commonly employed for biomarker detection rely on specific radio-, enzymatic-, or fluorescent-labeling to indicate the binding event. In contrast, there are innovative technologies known as label-free biosensors that eliminate the need for labeling, enabling the screening of complexes with minimal assay development [106]. While several label-free bone biosensors have been developed in recent years, none have become standard in clinical practice, as they were initially designed to assess a single marker. Caglar et al. introduced a microchip-based sensor for determining calcium ion levels. The technique involved measuring the reflectance index of arsenazo III immobilized on the surface of polymer beads [107,108]. A simple optical fiber and a ball lens were used to couple signals from a microchip to determine calcium ions in urine. Reflective mode increased assay sensitivity. Calcium ions interact with arsenazo III, forming a complex with a maximum absorption at 668 nm and a detection limit of 0.085 mM [108,109]. They noted that the microfluidic sensor demonstrated good agreement with other tools used for calcium ion measurement across a physiological range, emphasizing that its results were less influenced by competing ions in the samples.

The sensors discussed in this category have exhibited encouraging outcomes in quantifying the serum levels of specific biomarkers. Current research indicates that relying on a single bone turnover marker may not be adequate for identifying individuals at risk of fracture or effectively monitoring the treatment progress of osteoporosis. Hence, the development of a multiplex biosensor could offer significant value [22].

### 3.6. Biodegradable Biosensors

Researchers have also explored the concept of creating biodegradable iterations of biosensor devices to eliminate the necessity for subsequent surgery to extract the device from the healed fracture site. Sirivisoot et al. devised a biosensor incorporating a conductive, biodegradable polymer layer capable of sensing and regulating bone regrowth near the newly implanted orthopedic material [88]. The biosensor released bone growth factors to enhance bone formation, thereby facilitating tissue regeneration crucial for the success of the orthopedic implant. The degradation of the polypyrrole and poly-lactic-co-glycolic acid polymer layers, following sufficient bone growth, induced changes in conductivity that could be gauged by the CNTs extending from the anodized titanium on the sensor. This information was utilized to apprise the surgeon about the quantity of bone formed around the implant at different stages. In a separate study, Li et al. engineered a highly sensitive nano-field-effect transistor biosensor for investigating osteocyte mechanotransduction [110]. This innovative, label-free sensor, based on nanomanipulation, was designed to quantify the proteins released from mechanically stimulated osteocytes. Furthermore, a unique microbend optical biosensor, inspired by the Brain Neural Network, was developed to enhance biosensor performance in the early detection of osteoporosis [111]. This sensor was employed to detect and measure surface displacement, pressure, and strain on live bones.

## 4. Bone Physical Sensors

Bone physical sensors, also referred to as biomechanical sensors [107,112], detect and quantify various parameters like force, pressure, and temperature, often translating them into electrical resistance signals. However, they can be susceptible to nonspecific analyte interactions due to their sensitivity to environmental factors and temperature fluctuations [113,114,115]. Based on their working principles and operations, these sensors are categorized into electric, optical, piezoelectric, etc.

### 4.1. Electrical Sensors

Electrical sensors, known for their simplicity and widespread use, respond to physical stimuli by altering resistance, capacitance, or inductance. These sensors are categorized into resistive, capacitive, and inductive types based on their electronic components. Resistive sensors, such as strain gauges, detect variations in circuit resistance due to physical changes. Zhao et al. (2020) developed strain gauges, commonly attached to metal elastomer surfaces, which measure external-force-induced deformation by altering resistance [116]. Although strain gauges represent the benchmark for measuring bone strain in vivo, their application in humans is restricted due to the requirement for surgical implantation and the use of cyanoacrylate adhesives to affix the sensors to the bones. Fukase et al. (2022) created a wireless smart bone plate utilizing electrical impedance spectroscopy to monitor tissue composition during healing, operational wirelessly for up to 8 weeks and capable of readings within a 3 m range from the implanted bone [117]. Further, Wen et al. explored a microfabricated strain gauge suitable for application on living bones. In this study, a thin-film strain gauge was encapsulated within a poly-dimethyl-siloxane membrane. The objective of this study was to create an implantable sensor array capable of tracking surface strain on living bones. The research team employed an innovative approach to improve the mechanical durability of the sensor. Instead of using the traditional lift-off method for patterning the thin metal film, they opted for wet etching, resulting in the development of a miniaturized sensor. The outcomes demonstrated that the suggested strain gauge exhibited greater accuracy compared to commercially available alternatives [76]. In 2010, Umbrecht et al. introduced wireless implantable passive strain sensors (WIPSSs) for monitoring orthopedic implant deformities. These were constructed from biocompatible PMMA and filled with an incompressible fluid, WIPSSs utilized the amplification effect in hydromechanical systems for sensing. Achieving a strain resolution of 1.70 ± 0.2 × 10^−5^ and a dynamic input frequency range of 0.1–5 Hz, the sensor’s signal bandwidth extended up to 1 Hz due to decreased output with increased input frequency (Figure 3) [80].

### 4.2. Optical Biosensors

Researchers have increasingly focused on optical sensors, drawn to their rapid responsiveness, heightened sensitivity, real-time monitoring capabilities, immunity to electromagnetic interference, and biocompatibility [87,118]. Optical Bragg grating fibers are extensively utilized for measuring strain in various applications. These grating components are inscribed into the core of an optical fiber, serving as selective filters for light. They reflect spectral elements within the fiber core in accordance with the Bragg relation λ = 2 nΛ, where λ represents the wavelength, n denotes the mean reflective index of the core, and Λ signifies the spatial period of the refractive index modulation. If the fiber Bragg grating undergoes strain along the fiber axis, Λ changes. Consequently, the Bragg wavelength experiences a shift, serving as an indication of the degree of strain. Fresvig et al. suggested an alternative approach for monitoring bone deformation that involved the assessment of fiber optic Bragg grating sensors instead of traditional strain gauges. The evaluation encompassed measurements conducted on both an acrylic tube and a specimen of the human femur diaphysis. Four optical fiber sensors were employed for these measurements, with four strain gauges serving as a benchmark for result validation. However, no significant disparity was observed between the two measurement devices, whether in the measurement on the acrylic tube or the human bone sample [85,119].

### 4.3. Piezoelectric Sensors

The piezoelectric effect occurs when a piezoelectric material is deformed, leading to the generation of an electrical charge [74]. Piezoelectric sensors have demonstrated consistent outcomes in detecting alterations in the mechanical attributes of bone. Bender et al. detailed the application of piezoelectric sensors in tracking the formation of capsules around soft-tissue implants. Subsequently, a biosensor incorporating piezoelectric ceramic (PZT) was developed to assess the mechanical characteristics of bones. In this approach, two PZT patches were affixed to the bone, with one serving as the actuator and the other as the sensor. The bone was stimulated by applying an AC signal to the actuator, and the resulting variations were detected by the sensor patch. The biosensor created could evaluate alterations in the mechanical properties of bone through modifications in the frequency-response function (FRF). The method was proposed for monitoring the recuperation progress of bones following surgical procedures [81]. Hsieh et al. developed a contact-type piezoresistive micro-shear-stress sensor specifically for measuring the shear stress on a knee prosthesis. The sensor, based on a microelectromechanical system (MEMS), incorporated two transducers that converted stress into voltage. The sensor achieved a sensitivity of 0.13 mV/mA-MPa within a 1.4 N shear force range [82]. Additionally, Alfaro et al. introduced an ultra-miniature multiaxis implantable sensor designed to measure bone stress at a microscale level. Utilizing CMOS-MEMS technology, the device comprised an array of piezoresistive sensor pixels to monitor stress at the interface of the MEMS chip and bone [83].

## 5. Current Applications of Bone Sensors

As the population ages and the frequency of traumatic incidents increases, there is a noticeable shift towards utilizing implants to address damaged or deteriorating tissues within the body. In orthopedic contexts, certain implants are integrated with sensors for internal data collection and implant monitoring. Over recent years, a variety of multifunctional implants have emerged, enabling clinicians to remotely manage them via smart devices. Experts foresee these adaptable implants as catalysts for the next wave of technological advancements. While some of these sensors have received FDA approval, many are either undergoing clinical trials or preclinical animal studies. Despite growing research interest in implantable biosensors for musculoskeletal health, their practical application remains limited, with only a few products currently available on the market. The open eDisk, developed by Theken Disc, LLC (Akron, OH, USA), stands out as a noteworthy biophysical sensor in the market. It is crafted for tracking motion and loads within spinal disc implants. With its force-monitoring feature, it delivers personalized, real-time data to gauge a patient’s readiness for returning to work. Additionally, its microelectronic module sends wireless alerts to the wearer regarding any dynamic high-load events through an audible alert unit [120]. Similarly, the FDA granted de novo classification for Persona IQ, the sole FDA-cleared smart knee implant for total knee replacement. It records gait data, wirelessly transmitting to a home base station, then securely to surgeons via an HIPAA-compliant platform. Equipped with internal motion sensors, it facilitates post-surgery recovery monitoring for up to 20 years [121]. Numerous implantable biosensors are currently in preclinical and clinical stages, requiring additional validation to foster future translation into practical applications (Table 3).

## 6. Conclusions and Future Directions

This review article explores recent advancements in bone biosensing techniques and various biosensor types, placing particular emphasis on electrochemical biosensors for monitoring bone health due to their heightened sensitivity and reliability. Osteoporosis, often asymptomatic until a fracture occurs, underscores the importance of detecting BTBMs. These biomarkers, derived from bone-tissue enzymes or proteins, reflect various stages of bone remodeling and are measurable in blood or urine, offering insights into bone health. Detecting such biomarkers promptly is crucial for diagnosing osteoporosis, bone cancer, and infections, revealing underlying processes. These biomarkers, ranging from cells to gene products, aid in disease staging and treatment optimization. Accurate identification and monitoring of specific bone biomarkers are essential for refining diagnoses and optimizing therapeutic interventions for bone diseases. However, conventional methods for measuring bone mineral density are costly, time-consuming, and require specialized skills. Hence, there is a demand for portable electronic devices that can convert biological information into readable output. Yet, integrating them into biological environments poses challenges due to attachment complexities and signal processing. Unlike conventional methods, bone biomechanical sensors offer real-time assessments of bone biomechanics, providing immediate feedback on bone status. This capability sets them apart, offering timely detection of issues as they arise. However, it is crucial to acknowledge that existing sensors pose certain limitations and challenges for widespread applications. Consequently, there is a pressing need for further exploration and development of biosensors geared towards real-time monitoring, featuring simpler, faster, more cost-effective, and user-friendly approaches. Most current bone biosensors are designed to detect a single biomarker related to bone health. Yet, single biomarker detection proves inadequate for the precise and timely identification of bone diseases. Therefore, the development of methods for multiplex detection for multiple analytes is critical to address these challenges. Multiplex assays exhibit high sensitivity and necessitate less sample volume. Additionally, to effectively combat the growing concerns related to bone health, there is a requirement for advanced biosensors capable of detecting multiple markers through label-free detection techniques in the near future.

## Figures and Tables

**Figure 2 sensors-24-06172-f002:**
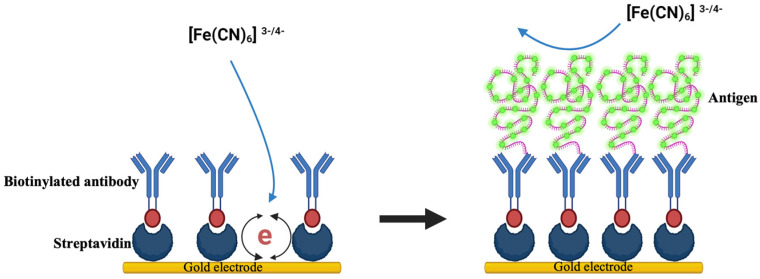
A schematic diagram illustrating a label-free immunobiosensor for detecting CTX-1, adapted and modified from reference [91].

**Figure 3 sensors-24-06172-f003:**
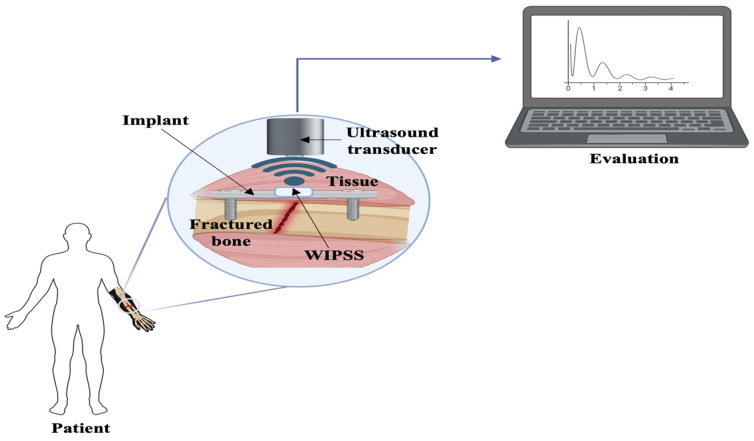
A schematic illustrating the concept of a wireless, implantable passive strain sensor, adapted and modified from reference [80].

**Table 1 sensors-24-06172-t001:** Commonly used analytical methods to measure bone turnover markers.

Marker	Secreted by/Source	Biological Sample	Analytical Method
**Bone Formation Biomarker**			
Bone-specific alkaline phosphatase (BALP)	Osteoblasts	Serum	Colorimetric, ELISA, ECLIA, RIA
Osteocalcin (OC)	Osteoblasts, odontoblasts	Serum	ELISA
C-terminal propeptide of type I procollagen (PICP)	Osteoblasts	Serum	ELISA, RIA
N-terminal propeptide of type I procollagen (PINP)	Osteoblasts	Serum	ELISA, ECLIA, RIA
**Bone Resorption Biomarker**			
Hydroxyproline (OHP)	Byproduct of collagen degradation	Urine	Colorimetric, HPLC
Hydroxylysine-glycosides (Hyl-glyc)	Byproduct of collagen degradation	Urine	HPLC, ELISA
Pyridinoline (PYD)	Formed during the extracellular maturation of collagen	Urine, Serum	HPLC, ELISA
Deoxypyridinoline (DPD)	Formed during the extracellular maturation of collagen	Urine, Serum	HPLC, ELISA
Amino-terminal cross-linked telopeptide of type I collagen (NTX-I)	Released by cathepsin K cleavage of bone collagen	Urine, Serum	ELISA, ECLIA, RIA
Carboxy-terminal cross-linked telopeptide of type I collagen (CTX-I)	Released by cathepsin K cleavage of bone collagen	Urine, Serum	ELISA, RIA
Bone sialoprotein (BSP)	Osteoblasts, osteoclasts, osteocytes, odontoblasts, and hypertrophic chondrocytes	Serum	ELISA, RIA
Tartrate-resistant acid phosphatase (TRAP)	Osteoclasts, neurons, and activated macrophages	Serum	Colorimetric, ELISA, RIA
Cathepsin K (Ctsk)	Osteoclasts	Serum	ELISA

**Table 2 sensors-24-06172-t002:** Some of the existing physical sensors and biosensors for bone health monitoring.

Physical Sensors and Biomechanical Sensors			
Sensor	Principle	Parameter Measured	References
MPACT 3500 Project	Resistance changes proportional to strain	Implant strain	[75]
Microfabricated Strain Gauge	Changes in electrical resistance due to strain	Surface strain on live bone	[76]
Flexible Strain-gauge Sensor	Changes in electrical resistance due to pressure, shear, and torsion	Pressure, shear, torsion	[77]
Nanotube Film Strain-sensing System	Voltage across film changes linearly with strain	Multidirectional strain sensing	[78]
Ultrasound-based Wireless Implantable Passive Strain Sensor (WIPSS)	Hydromechanical effects	Deformation of implants	[79,80]
Piezoelectric sensor	Changes in frequency-response function (FRF)	Mechanical parameters of bones	[81]
Piezoresistive Micro-shear-stress Sensor	Transformation of stress into voltage	Shear stress of knee prosthesis	[82]
Ultra-miniature Multiaxis Implantable Sensor	Changes in resistance	Bone stress at microscale level	[83,84]
Fiber Bragg Grating Sensors	Changes in Bragg wavelength due to strain	Strain measurement	[85,86]
Photometric sensor	Microbending technique to measure bone strength	Bone strength	[87]
Biodegradable sensor	Measures conductivity variations as new bone forms	Monitoring orthopedic tissue growth	[88]
**Biosensors**			
Molecularly imprinted polymer biosensor for CTX-I	Selective binding of CTX-I molecules to synthesized antibodies	CTX-I	[89]
Carbon nanotube (CNT) electrodes coated with gold nanoparticles	Detection of CTX through antigen-antibody binding events on surface	CTX-I	[90]
Label-free immunosensor for C-terminal telopeptide bone turnover marker	Streptavidin immobilization, antibody binding, EIS for detection	CTX-I	[91]
Biosensor targeting osteocalcin	Covalent immobilization of antiosteocalcin antibody on gold electrode	Osteocalcin	[92]
Electrochemical biosensor for ALP determination	Disposable graphite screen-printed electrodes, SWV for quantification	ALP	[93]
Printed electrochemical biosensors for astronaut point-of-care testing	NTX antibodies are incorporated into the sensor’s electrode ink and read by handheld electronics for rapid measurements	NTX	[94]
Colorimetric sensor for alkaline phosphatase (ALP)	Detection of ALP activity through color change induced by AgNP growth	ALP	[95]
Novel NIR fluorescent probes for highly sensitive ALP detection	ALP cleaves the phosphate group in NIR probes, leading to a significant fluorescent signal increase	ALP	[96]
Selective, smartphone-based approach for visually detecting ALP using NH_2_-Cu-MOFs	Utilization of NH_2_-Cu-MOFs with oxidase mimicry and fluorescence capabilities for ALP detection	ALP	[97]
Automated multiplex immunoassay for bone turnover markers	Simultaneous measurement of CTX-I, PINP, OC, and PTH in 20 μL of serum	CTX-I, PINP, OC, PTH	[97]
Osteokit multiplex assay for bone marker assessment	Simultaneous measurement of OC and CTX-I in serum using a microfluidic platform	OC, CTX-I	[98]
Microchip-based sensor for determining calcium ion levels	Measurement of reflectance index of immobilized arsenazo III on polymer beads	Calcium ions	[97]

**Table 3 sensors-24-06172-t003:** Bone sensors currently in preclinical and clinical studies.

Sensor Type	Study	Implant	Remarks	Reference
Resistive sensors	Clinical studies	Hip	Sensors were integrated into a hip endoprosthesis to monitor joint contact forces and temperature distribution across the entirety of the titanium implant.	[122]
		Knee	An electronic knee prosthesis was surgically implanted to assess tibial forces in vivo during daily activities following total knee arthroplasty (TKA).	[123]
		Knee	To assess intercompartmental load intraoperatively following conventional gap balancing with a tensiometer during TKA.	[124]
	Preclinical animal studies	Spine	In two baboons, interbody implants with instrumentation were surgically implanted into the disc space of a motion segment to monitor the in vivo loads in the lumbar spine.	[125]
		Spine	Temperature-sensing implants could consistently identify local temperature fluctuations linked to peri-implant wound infections.	[126]
		Fracture	An implanted sensor system monitors implant load continuously to assess the status of bone healing.	[127]
		Fracture	A wireless, biocompatible microelectromechanical system sensor was developed, evaluated, and implemented in a large animal model for monitoring purposes.	[128]
Optical sensors		Spine	Records the intradiscal pressure signal from a sedated sheep while it’s spontaneously breathing.	[129]

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
