# Peer review of "Sensors in Bone: Technologies, Applications, and Future Directions"

_sensors, 2024, doi:10.3390/s24196172_

Round 1

Reviewer 1 Report

Comments and Suggestions for Authors

I think the review article is very interestingly written and covers a wide range of scientific papers. I have no negative comments about the article.

• What is the main question addressed by the research?

This review article discusses the latest advances in the development of electrochemical biosensors for bone health monitoring. In addition, it provides detailed information on the scientific advances in this area of research. Comparisons with other types of sensors for bone health monitoring are also provided.

• Do you consider the topic original or relevant to the field?

Yes

Does it  address a specific gap in the field?  

Yes

Please also explain why this is/ is not the case.

I believe that review articles are necessary for understanding and analyzing already published experimental data.

• What does it add to the subject area compared with other published material?

This review article complements the existing review articles. The authors express a well-founded point of view on how electrochemical sensors are better in comparison with other sensors.

• What specific improvements should the authors consider regarding the methodology? What further controls should be considered?

I believe that the article can be recommended for publication in its current form.
• Are the conclusions consistent with the evidence and arguments presented and do they address the main question posed?

Yes

 Please also explain why this is/is not the case.

I believe that the conclusions presented in the article correspond to the main content of the article.
• Are the references appropriate?

Yes

• Any additional comments on the tables and figures.

Perhaps more illustrative material could have been provided. But this remark does not reduce my high assessment of the presented article.

Author Response

Thank you very much for your positive and encouraging comments. I appreciate your thoughtful feedback and am glad you found the review comprehensive and valuable. Your support motivates me to continue contributing to this field.

Reviewer 2 Report

Comments and Suggestions for Authors

This review provided a comprehensive overview of the current state of sensor technologies used in bone, covering their integration with bone tissue, various applications, recent advancements, challenges, and future directions. This review article explores recent advancements in bone biosensing techniques and various biosensor types, placing particular emphasis on electrochemical biosensors for monitoring bone health due to their heightened sensitivity and reliability. The article concludes by pointing out there is a requirement for advanced biosensors capable of detecting multiple markers through label-free detection techniques in the near future.

Comments to the Author

1, In line 32, please indicate the reference number in square brackets. The labeling method for all subsequent references is also the same.

2, In line 91, please mark the abbreviation where it first appears, refer to line 82.

3, In line 97 and 98, please pay attention to the position of abbreviations and determine whether they appear in the title or after the corresponding specified word in the text. This issue also exists in subsequent articles, please pay attention to correcting it.

4, In line 251, whether Figure 1 is the author's research result or a cited literature should be clearly described and the citation source should be provided.

5, In line 414, Figure 4 is mentioned, but it was not found in the main text. Please provide an explanation.

Author Response

Thank you for taking the time to review my paper and for providing valuable suggestions. Your insightful feedback has been extremely helpful in improving the quality of the manuscript. I truly appreciate your effort and thoughtful recommendations.

1. In line 32, please indicate the reference number in square brackets. The labeling method for all subsequent references is also the same.

Thank you for your suggestion. I have made the necessary changes in line 32 by indicating the reference number in square brackets, and I have applied the same labeling method consistently throughout the manuscript

2. In line 91, please mark the abbreviation where it first appears, refer to line 82.

Thank you for pointing that out. I have updated line 91 to mark the abbreviation upon its first appearance, as noted in line 82

3, In line 97 and 98, please pay attention to the position of abbreviations and determine whether they appear in the title or after the corresponding specified word in the text. This issue also exists in subsequent articles, please pay attention to correcting it.

Thank you for your feedback. I have addressed the issue in lines 97 and 98 by ensuring that abbreviations are positioned correctly, either in the title or immediately after the corresponding specified word. I have also made sure to apply this correction throughout the entire paper

4, In line 251, whether Figure 1 is the author's research result or a cited literature should be clearly described and the citation source should be provided.

Thank you for your observation. Figure 1 has been adapted and modified from various sources. I have included the relevant references to these papers in the manuscript to indicate the origin of the figure and ensure proper citation

5, In line 414, Figure 4 is mentioned, but it was not found in the main text. Please provide an explanation.

Thank you for pointing that out. The mention of Figure 4 in line 414 was a typographical error. I have corrected this mistake.